# Environmental Effects of Commodity Trade vs. Service Trade in Developing Countries

Mohammad Zohaib Saeed and Shankar Ghimire *

School of Accounting, Finance, Economics and Decision Sciences, Western Illinois University, 1 University Circle, Macomb, IL 61455, USA
* Correspondence: sp-ghimire@wiu.edu; Tel.: +1-309-298-1152

**Abstract:** Increasing levels of carbon emissions have been a growing concern worldwide because of their adverse environmental effects. In that context, this paper examines the association between different categories of trade and carbon dioxide emissions. In particular, we analyze whether total trade, commodity trade, and service trade affect the environment differently. The analysis is based on panel data for 147 developing countries for the period from 1960 to 2020. Methodologically, the fixed-effects model, as suggested by the Hausman test, is used to examine the relationships. We present two main conclusions: (1) overall trade increases $CO_2$ emissions, and (2) commodity trade contributes to higher levels of $CO_2$ emissions than service trade. These results have important policy implications—climate change policies should target commodity trade sectors to help reduce environmental carbon emissions.

**Keywords:** carbon dioxide emission; trade liberalization; environment; commodity trade

## 1. Introduction

Globalization has enabled countries to easily consume and exchange various goods and services beyond their borders. Trade is more open now than ever before because of the removal of barriers such as tariffs, quotas, etc., along with other liberal trade policies that countries have adopted to support their economic growth [1]. Trade liberalization allows countries to fully or partially remove certain types of international trade barriers, either in the form of tariffs or non-tariff barriers [2], enabling nations to focus on producing goods and services in which they have a comparative advantage [3]. That is, what a country trades depends on the opportunity cost, with lower opportunity costs resulting in improved efficiency [4,5]. In this process, a country could either help the environment by producing environmentally friendly products or hurt the environment by overproducing goods with negative externalities.

The impact of trade liberalization on the environment is a controversial issue. Some studies show that trade liberalization increases $CO_2$ emissions that degrade the environment [6–8], while other empirical evidence states the opposite [9,10]. Trade results from the production of goods and services that increase $CO_2$ emissions, but it is difficult to quantify the extent to which individual goods and services contribute to carbon emissions. For instance, the agriculture, manufacturing, mining, and service industries are different in nature and could affect the environment at different levels. Scant literature examines how various trade categories impact the environment at the national level [10–12]. A key issue with the single-country studies in the existing literature is that greenhouse gas (GHG) emissions are a global problem, i.e., an international externality. An individual country cannot do much about carbon emissions on its own, due to the free-rider problem [13]; hence, a broader analysis at the global level is necessary to understand such a global phenomenon.

Due to the ambiguities in the current literature about the trade–environment nexus and the limited scope of the analysis, this paper uses a global dataset to further examine how

different trade structures increase $CO_2$ gas emissions and negatively affect the environment. In particular, we analyze the disaggregated effects of total trade (percentage of GDP), commodity trade (using merchandise trade as a percentage of GDP as a proxy), and service trade (percentage of GDP) on $CO_2$ emissions. In doing so, we fill the gap in the literature by providing empirical evidence that commodity trade and service trade impact the environment differently across all developing countries.

Disaggregating trade is vital to study how different sectors impact the environment in developing countries, because some developed countries outsource and/or offshore certain types of commodity production to developing countries for lower costs and less-stringent environmental policies [14]. As a result, more pollution is emitted in developing countries with lax environmental policies [15] than in developed countries. As such, developing countries are more vulnerable to environmental pollution than developed countries [16] if the pollution haven hypothesis holds [17,18], although it is possible to see the pollution halo hypothesis and improve environmental quality [19] if the use of environmentally friendly and clean technology is promoted in the production process. Existing literature on the trade–environment nexus focuses on total trade [9,12,20,21]. However, the analysis focusing on the impact of total trade on the environment underestimates the effects of trade categories with high negative impacts while overestimating the effects of other types with less harmful impacts. Hence, this study aims to investigate the environmental impacts of different trade categories in developing countries to show which trade categories adversely affect the environment.

This study contributes to the climate change literature by disaggregating total trade into commodities trade (using merchandise trade as a proxy) and service trade and estimating their role in carbon emissions—a proxy for environmental quality. Merchandise trade is defined as "goods which add or subtract from the stock of material resources of a country by entering (imports) or leaving (exports) its economic territory" [22]. Since merchandise trade includes most physical goods from agriculture, industry, manufacturing, etc., we use it as a proxy for commodity trade. In contrast, service trade includes services in information and communication, construction, distribution, educational, energy, environmental, financial, health, social, tourism, transport, business, and professional services [23]. Based on how commodity and service trade are measured, we hypothesized that commodity trade impacts the environment adversely, while service trade is less detrimental. Our results support these hypotheses. Furthermore, the results have important policy implications, as discussed by environmentalists, climate activists, and policymakers.

The rest of this paper is organized as follows: Section 2 examines related literature; Section 3 presents the data and methodology; Section 4 discusses the empirical results; and Section 5 concludes with a summary and policy implications.

## 2. Literature Review

This section presents an overview of the existing literature on how different types of trade impact the environment. We also present a summary of the literature highlighting the overall impact of trade liberalization on the environment. There is a vast body of literature on both theoretical and empirical analysis for estimating the effects of trade liberalization on the environment, using carbon dioxide as a proxy for environmental quality due to its highest share in greenhouse gas emissions [24]. Table 1 provides a summary of the representative literature.

In addition to the sectoral analyses listed in Table 1, the literature finds both positive and negative impacts of trade liberalization on the environment, mainly focusing on $CO_2$ emissions. A study revealed that trade liberalization hurts the environment by increasing $CO_2$ as the most-emitted and $N_2O$ as the least-emitted GHGs in the atmosphere [20]. This study also argues that the proportion of $CO_2$ in the GHGs is 90%. Another study used $CO_2$ emissions as the proxy for environmental quality with fixed- and random-effects methods on panel data from 2002 to 2016 [21]. The results showed a negative association between the environmental performance index and trade in the selected countries.

**Table 1.** Review of literature for agriculture, manufacturing, and service trade.

| Title and Author(s) | Objectives | Results and Findings |
| --- | --- | --- |
| *Trade in Agriculture* | | |
| Environmental effects of agricultural trade liberalization [25]. | Examined environmental impacts due to agricultural trade liberalization and domestic policy reforms through existing literature. | Indirect impacts were caused due to changes in location, intensity, product mix, and agricultural technology, while direct impacts included pollution due to the transportation of agricultural products, as well as the potential migration of harmful species of plants, animals, and insects that could disrupt the food chain. |
| The environmental impacts of agricultural trade: A systematic literature review [26]. | Investigating the relationships between agricultural trade and the environment based on existing literature in international economics. | Focused on local (land use) rather than global emissions (GHGs). A negative relationship between agricultural trade and the environment (pollution) was found, but some studies show a positive relationship, while very few studies could not find any relationship. |
| Considering the full array of impacts on human health and the environment by products, services and supply chain [27]: US Environmental Protection Agency Report. | Manufacturing trade-offs in the environment, health, and resource use. | Selection of bioproducts by the producers reduces fossil fuel extraction and inputs as well as GHGs, but the growing and harvesting of bio-feedstocks may also pollute water sources and degrade soil quality. |
| The role of exports in manufacturing pollution in sub-Saharan Africa, South Asia: toward better trade-environment governance [28]:UNCTAD Report (2021). | To assess the export-associated pollution by the manufacturing sector in sub-Saharan Africa and South Asia. | Identified the public and private environmental governance opportunities to achieve more sustainable production linked to trade. Should follow sustainable practices despite adding value in the products in the manufacturing sector. |
| *Trade in Services* | | |
| Does Service Trade Globalization Promote Trade and Low-Carbon Globalization? Evidence from 30 Countries [29]. | Investigated the effects of service trade globalization on low-carbon globalization through the Unified Efficiency Index and Energy-Environmental Performance Index in 30 countries from 1980 to 2013 using a tobit model. | Service trade openness showed positive effects on both energy and $CO_2$ emission efficiency that have further improved with time, which is good for the environment. Emerging service sectors promoted the improvement of energy and $CO_2$ emission efficiency, while the traditional sectors hindered the efficiency improvement. There existed a "catch-up" effect between less-developed countries and developed countries in terms of energy and $CO_2$ emission efficiency. |
| Does trade in services improve carbon efficiency?—Analysis based on international panel data [30]. | Finding the theoretical and empirical impacts of service trade on carbon efficiency using panel data for 55 countries from 2001 to 2015 using a slacks-based measure model with a global Malmquist–Luenberger index. | Carbon efficiency was improved due to service trade, which was greater in exports than imports. Each country showed a different effect for exports and imports of services. |

A similar study examined the associations between decomposed goods and services and $CO_2$ and $SO_2$ emissions using panel data for 179 countries segregated by the Organization for Economic Cooperation and Development (OECD) and non-OECD countries. Surprisingly, trade in goods reduced the $CO_2$ intensity for OECD countries over 20 years of data when there was a boom in international trade. However, no significant impact of trade in services was reported. Moreover, OECD countries involved in international trade used clean technology to produce goods [9]. Another study decomposed merchandise trade into imports and exports for some developing countries between 1971 and 2013 [31]. The results showed that imports increased $CO_2$ emissions, but exports and economic growth could improve the environment.

At the individual country level, a study examined the issues in China and found different effects of trade liberalization on the environment using a structural decomposition analysis on various tradable goods. It showed that the scale effect resulted in most carbon emissions being partially offset by the positive technique effect. However, the composition effect had no significant impact, resulting in overall environmental degradation [12]. The study also used $CO_2$ as a proxy for environmental quality and found that trade liberalization deteriorates the environment. Similarly, another study in China used $CO_2$ and $SO_2$ as a proxy for environmental quality. Firm-level panel data from the period 1998–2007 were collected from the manufacturing sectors, along with data on pollution emissions and

the degree of trade liberalization. Despite the positive impact of the technique effect, the combined negative impact of scale and composition offset this positive impact, resulting in overall environmental degradation [11].

In sum, the existing literature shows an inverse relationship between trade liberalization and the environment, using $CO_2$ as a proxy for environmental quality. On the other hand, the literature also shows a positive effect of trade liberalization on the environment, but only in the short term [32]. We studied this issue further by analyzing the disaggregated effects of various components of trade, mainly focusing on commodity and service trade. In particular, we tested the following two hypotheses:

1. $CO_2$ emissions are positively associated with trade volume.
2. Commodity trade contributes to higher levels of $CO_2$ emissions than service trade.

## 3. Methodology

### 3.1. Empirical Model

We ran the three regression models represented by Equations (1)–(3) to test the hypotheses presented above.

$$logCO_{2it} = \beta_0 + \beta_1 TTR_{it} + \beta_2 GDPGrowth_{it} + \beta_3 FDI_{it} + \beta_4 logK_{it} + \beta_5 LFP_{it} + \alpha_i + \varepsilon_{it} \quad (1)$$

$$logCO_{2it} = \beta_0 + \beta_1 CTRr_{it} + \beta_2 GDPGrowth_{it} + \beta_3 FDI_{it} + \beta_4 logK_{it} + \beta_5 LFP_{it} + \alpha_i + \varepsilon_{it} \quad (2)$$

$$logCO_{2it} = \beta_0 + \beta_1 STR_{it} + \beta_2 GDPGrowth_{it} + \beta_3 FDI_{it} + \beta_4 logK_{it} + \beta_5 LFP_{it} + \alpha_i + \varepsilon_{it} \quad (3)$$

The model specification follows the existing literature examining the determinants of environmental quality [20,32,33]. Our key outcome variable is the log of $CO_2$ gas emissions—an environmental quality indicator. Total trade (TTR), commodity trade (CTR), and service trade (STR) are the key variables of interest included in Equations (1), (2), and (3), respectively. We also control for variables that the literature shows to be determinants of carbon emissions, i.e., annual GDP growth (GDPGrowth), foreign direct investment as a percentage of GDP (FDI), natural log of capital stock (logK), and the labor force participation rate for those aged 15 years and older (LFP). $CO_2$ emissions are measured in kilotons and the capital stock (i.e., gross fixed capital formation) is measured in USD values. We take the natural log of $CO_2$ emissions and the capital stock to linearize the regression model. Other control variables are expressed either in ratios or percentage forms, and no further transformations are performed. Table 2 provides further details of each variable.

**Table 2.** Description of variables.

| Variable | Description | Unit of Measurement |
|---|---|---|
| $logCO_2$ | Log of carbon dioxide emissions | Kilotons |
| TTR | Total trade | As a percentage of GDP |
| CTR | Commodity trade | As a percentage of GDP |
| STR | Service trade | As a percentage of GDP |
| GDP | GDP growth | Annual percentage |
| FDI | Foreign direct investment | Net inflows as percentage of GDP |
| logK | Log of capital | Gross fixed capital formation |
| LFP | Labor force participation rate | Percentage of total population aged |
| $\alpha_i$ | Individual country-specific effects | 15+ |
| $\varepsilon$ | Error term | |
| *it* | Country and time period | |

Other notations in the regression model have their usual meaning—$\beta_i$ represents the parameters of interest, $\alpha_i$ indicates the individual country-specific effects, and $\varepsilon_{it}$ is the white noise for country *i* at time *t*. Consistent with our testable hypotheses, the three $\beta_1$ coefficients having positive estimates would mean that higher trade volumes increase $CO_2$ emissions, while negative estimates would suggest the opposite.

*3.2. Data Description*

To compute the relationship between trade liberalization and environmental quality by segregating total trade into commodity versus service trade, we used $CO_2$ emissions as a dependent variable, measured in kilotons, as used widely in the existing literature [9,31,34]. According to the World Development Indicators database, carbon dioxide emissions mostly consist of byproducts of the production and use of energy, including the burning of fossil fuels (i.e., their use in agriculture, manufacturing, industry, and other human activities). The measure excludes emissions from land use, e.g., deforestation. Estimates also exclude fuels supplied to ships and aircraft for international transport, because of the difficulty of apportioning the fuels among benefiting countries.

We had three key explanatory variables: total trade as a percentage of GDP was used for TTR; merchandise trade as a percentage of GDP was used for CTR; and service trade as a percentage of GDP was used for STR. Other control variables included in the model were FDI (net inflows as a percentage of GDP), GDP growth (annual percentage), log of capital (gross fixed capital formation in USD, 2015), and labor force participation rate (percentage of the total population aged 15+). We controlled for these variables because they affect environmental quality [21]. A study in the South Asian Association for Regional Cooperation (SAARC) countries found that capital and economic growth negatively impact environmental quality in the long term. Another study showed that FDI, capital, and economic growth positively impact the environment in the short term [32]. Moreover, the existing literature on FDI shows that it can support either the pollution haven hypothesis and increased $CO_2$ [17,18] or the pollution halo hypothesis and decreased $CO_2$ [19], depending on the sample. The literature shows that the population is also a key determinant of carbon emissions [35]. In particular, the population age and structure affect carbon emissions, primarily through the expansion of the labor force and subsequent overall economic growth [36]. To capture this effect, we included the labor force participation rate (i.e., the percentage of the total population aged 15+) in the model.

Our panel data for a sample of 147 developing countries were obtained from the World Development Indicators database maintained by the World Bank. The list of developing countries was obtained from the International Monetary Fund (IMF) website, which divides countries into two groups: advanced economies, and developing and emerging economies. According to the IMF, the main criteria used for country classification are (i) per capita income level, (ii) export diversification, and (iii) degree of integration into the global financial system. The IMF uses either sums or weighted averages of data for individual countries. However, the IMF's statistical appendix for country classification explains that this is not a strict criterion, and other factors are considered in deciding the classification of countries. The World Bank [37] also designates nations and territories with a GNI of USD 12,535 or less as low-income economies, most of which are on the list of developing countries per the IMF classification. Based on this criterion and the availability of data in the World Bank database, the developing countries selected for this study comprised 147 out of the total 156, as listed in Appendix A. Table 3 shows the data summary.

**Table 3.** Descriptive statistics.

| | Mean | Standard Deviation | Minimum | Maximum | Count |
|---|---|---|---|---|---|
| Log of $CO_2$ | 3.5239 | 1.1509 | 0.5643 | 7.0134 | 7821 |
| Total trade | 69.3250 | 37.3352 | 0.0210 | 245.3694 | 6063 |
| Commodity trade | 55.7831 | 32.3571 | 2.7226 | 311.5114 | 7056 |
| Service trade | 21.3543 | 18.8302 | 0.6250 | 143.9805 | 5121 |
| GDP growth | 3.7742 | 6.9498 | −64.0471 | 149.9730 | 6870 |
| Foreign direct investment | 3.0166 | 5.8095 | −55.2341 | 161.8238 | 6172 |
| Labor force participation rate | 62.0268 | 12.1110 | 31.4400 | 90.3200 | 4185 |
| Log of capital | 9.6331 | 0.8879 | 5.7481 | 11.9131 | 4037 |

Notes: (1) there are many missing observations, which is why the total count for each variable is different; (2) we removed the outliers before running the final regression analysis.

*3.3. Estimation Technique*

We began the analysis with the standard fixed- and random-effects methods for panel data analysis presented above, as described in the literature [21,34]. To identify the appropriate estimation method for the given panel data, we used the Hausman specification test [38] after regressing each model using Equations (1)–(3). The test identifies whether the individual country-specific effects $\alpha_i$ are correlated with the covariates ($X_{it}$).

$$H_0 : Corr(\alpha_i, \ X_i) = 0$$

$$H_1 : \ Corr(\alpha_i, \ X_i) \neq \ 0$$

If the null hypothesis is true, it indicates that the random-effects model would be appropriate for the data. However, if the alternative hypothesis is true, the fixed-effects model would be appropriate for the data. Table 4 presents the results from the Hausman specification test for each model represented in Equations (1)–(3).

**Table 4.** Hausman test results.

| Model | Chi-Squared Statistic | *p*-Value |
|---|---|---|
| Total trade as a percentage of GDP (TTR) | 34.26 | <0.0001 |
| Commodity trade as a percentage of GDP (CTR) | 62.722 | <0.0001 |
| Service trade as a percentage of GDP (STR) | 63.973 | <0.0001 |

For each model, the *p*-value is smaller than 0.05. Thus, we can reject the null hypothesis of random effects as an appropriate estimation method; that is, we use the results from the alternative hypothesis—the fixed effects—to examine our testable hypotheses. Intuitively, the fixed-effects model makes sense, as certain cross-country differences do not change over time, and it is necessary to control for them. These models were estimated with R software using the *plm* package.

## 4. Empirical Results and Discussion

This section presents the regression results comparing and contrasting the impacts of various trade categories on environmental quality.

*4.1. Results Pertaining to Key Variables of Interest*

The results obtained from estimating Equations (1)–(3) using the fixed-effects method are shown in Table 5. Each column represents the result for each model. The dependent variable is the natural log of carbon dioxide emissions, and our main explanatory variables are total trade (percentage of GDP), commodity trade (merchandise trade, percentage of GDP, as a proxy), and service trade (percentage of GDP). We can see that total trade has a positive and statistically significant association with carbon dioxide. Specifically, a 1% increase in trade increases carbon dioxide by 0.3%, which reflects a moderate increase in environmental pollution. This further illustrates that carbon dioxide emissions also increase as the total trade increases, adversely impacting environmental quality. These results are consistent with the existing literature [6,7,20,21,32]. This supports our first testable hypothesis.

We also see similar results for commodity trade, i.e., positive and significant relationship with carbon dioxide. As commodity trade increases by 1%, it increases $CO_2$ by 0.4%, which reflects that commodity trade also results in deteriorating environmental quality at a large scale. While these results are consistent with the existing literature, one study showed a contradictory result, where trade in commodities reduced the $CO_2$ emissions and improved the environment in OECD countries [9]. A possible reason for this is the increase in international trade due to technological improvement, especially in developing countries. Furthermore, service trade did not show a significant impact, as expected. There was a positive but statistically insignificant relationship between service trade and carbon

dioxide emissions. Existing evidence shows similar results, with an insignificant impact of service trade on $CO_2$ emissions [9]. The results indicate that most of the negative effects of trade on the environment come from commodity trade, rather than from service trade. This means that an increase in service trade does not directly increase carbon dioxide. This makes sense based on the definition of service trade as trade in services only, which are non-physical in nature. Hence, service trade does not deteriorate the environment, even if it does not contribute to improving the environmental quality. These results are consistent with the existing literature; unlike commodity trade, which is potentially harmful to the environment due to the increased output of tangible goods, trade in services is less harmful to the environment [29,30]. These results support our second testable hypothesis.

**Table 5.** Results obtained from the fixed-effects estimation.

| | Dependent Variable: | | |
|---|---|---|---|
| | $LogCO_2$ | $LogCO_2$ | $LogCO_2$ |
| Total trade (% of GDP) | 0.003 *** | | |
| | (0.0002) | | |
| Commodity trade (% of GDP) | | 0.004 *** | |
| | | (0.0003) | |
| Service trade (% of GDP) | | | 0.001 |
| | | | (0.001) |
| GDP growth rate | −0.010 *** | −0.011 *** | −0.010 *** |
| | (0.002) | (0.002) | (0.002) |
| FDI (% of GDP) | −0.012 *** | −0.009 *** | −0.002 |
| | (0.002) | (0.002) | (0.002) |
| Log capital | 1.024 *** | 1.011 *** | 0.984 *** |
| | (0.009) | (0.009) | (0.010) |
| Labor force participation rate | −0.008 *** | −0.008 *** | −0.009 *** |
| | (0.001) | (0.001) | (0.001) |
| Observations | 2580 | 2602 | 2397 |
| $R^2$ | 0.857 | 0.856 | 0.844 |
| Adjusted $R^2$ | 0.855 | 0.854 | 0.842 |
| F-statistic | 3057.205 *** | 3051.155 *** | 2560.033 *** |
| | (df = 5; 2546) | (df = 5; 2568) | (df = 5; 2363) |

Note: *** $p < 0.01$.

*4.2. Results Pertaining to Control Variables*

The overall results presented in Table 4 show a significant result based on the values of $R^2$ and adjusted $R^2$ (approximately 85%). The F-statistic also shows a highly significant value for all of the models. Most of the independent variables also show a high correlation with the dependent variables, making the overall estimation robust.

GDP growth shows a negative association with carbon dioxide. This further highlights that as GDP growth increases, carbon dioxide decreases, improving the environmental quality. The results are consistent with the literature on the Kuznets curve hypothesis, which suggests that environmental pollution increases at the beginning of economic growth, but when it passes a certain level of income the economic growth helps to improve environmental quality [39].

FDI shows a negative and significant relationship with carbon dioxide emissions when we consider total trade and commodity trade, but the relationship is statistically insignificant when service trade is considered. Existing literature shows that FDI can increase $CO_2$ emissions based on the pollution haven hypothesis [20] or decrease them based on the pollution halo hypothesis [19]. Based on our results, it is possible that the foreign investments are used to move towards the use of environmentally friendly and clean technology in the production process, which would improve the environmental

quality [34]. However, these findings require a separate investigation that is outside the focus of this paper, because the effects of FDI could be different based on the stringency of the receiving countries' environmental policies [40,41] and the income levels of the countries [42,43], while the effect could also be bidirectional [44] and depend on the time horizon [33].

Capital stock also plays an important role in determining carbon dioxide emissions. The coefficients of log-capital are consistently positive and statistically significant across all specifications, indicating a significant association with carbon emissions. A potential explanation for this observation is the positive role of capital in increasing output and economic activity [45,46], which would ultimately increase $CO_2$ emissions in the initial stages of growth. On the other hand, the labor force participation rate negatively and significantly affects carbon emissions. However, these results are inconsistent with the existing literature [36], potentially due to differences in the samples of countries included in the analysis. There is limited literature examining the role of variables such as capital and labor, and a detailed study on this topic is necessary to understand the inconsistent results.

By pooling the results from the three models, the data provide statistical evidence that both the total trade and commodity trade have a positive and statistically significant relationship with carbon dioxide emissions. However, service trade does not have a significant relationship with $CO_2$ emissions. This implies that higher commodity trade will increase carbon dioxide emissions and deplete environmental quality.

### 4.3. Study Limitations and Future Prospects

This paper analyzed the impacts of commodity versus service trade on environmental outcomes. While disaggregating total trade into commodity and service trade provides us with a way to understand the issue further, there are some limitations. It would be ideal to analyze the impacts of other disaggregated trade components, such as hard commodities and soft commodities, including agriculture, manufacturing, mines and minerals, etc. This would provide a better picture to find and compare the major sources that release the most carbon dioxide in the atmosphere. Additionally, using the firm-level data for a few developing countries would allow us to empirically analyze the decomposition of trade commodities to capture the in-depth impacts of trade. Detailed analysis of the individual commodities is beyond the scope of this paper. Furthermore, many countries' economic structures have been changing since the COVID-19 pandemic [47,48]. How the pandemic affects their economic structures and, ultimately, their carbon emissions will be a contentious issue. Future research should focus on these issues.

### 5. Summary, Conclusions, and Policy Implications

This paper's main objective was to find and distinguish the impacts of international trade on the environmental quality of developing countries, which are more susceptible to climate change. The goal was to see whether segregating trade into commodity and service trade would affect the environmental quality differently. We postulated two testable hypotheses: (1) overall trade increases $CO_2$ emissions, and (2) commodity trade contributes to higher levels of $CO_2$ emissions than service trade. We tested these hypotheses by examining the effects of total trade (as a percentage of GDP), commodity trade (using merchandise trade as a percentage of GDP as a proxy), and service trade (as a percentage of GDP) on $CO_2$ emissions (a proxy for environmental quality). We used panel data from 147 developing countries for the period 1960–2020 and analyzed the data using the fixed-effects model as suggested by the Hausman test. We estimated three models separately to compare and contrast the effects of different trade categories. The results show that an increase in total trade negatively impacts the environment, mainly contributed by commodity trade rather than service trade. Thus, it can be inferred from the results that commodities release a higher share of carbon dioxide emissions in contrast to trade in services. Commodity trade involves tangible goods such as manufacturing, agriculture, mines and minerals, stones and precious elements, etc., which emit several harmful gasses as byproducts during

their production processes. On the other hand, service trade is trade in services such as human resources and skills in information technology, engineering, etc., showing no statistically significant effect on the environment. Hence, our results support both of our testable hypotheses.

From a policy perspective, appropriate tools are necessary to control harmful gas emissions from the manufacturing and industrial sectors, which make up most of the trade in commodities. To do so, it is imperative to first identify the commodities produced by developing countries. One way to curb carbon emissions is to use environmentally friendly technology to produce hard commodities such as metals and iron and to extract elements such as coal, chromite, and minerals. Governments could also play an appropriate role in providing subsidies to invest more in environmentally friendly commodities through environment and climate change ministries in collaboration with international NGOs. Moreover, countries could devise a carbon pricing policy to either increase the tax on each unit of carbon emission generated or promote the production of commodities with lower carbon emissions. According to world trade figures, countries such as China, India, Russia, etc., trade the highest amounts of commodities. Therefore, they should act immediately and pass on the appropriate strategies to other countries to help them reduce carbon emissions and improve environmental quality.

**Author Contributions:** Conceptualization, S.G. and M.Z.S.; methodology, S.G. and M.Z.S.; software, M.Z.S.; validation, S.G. and M.Z.S.; formal analysis, S.G. and M.Z.S.; investigation, S.G. and M.Z.S.; resources, S.G. and M.Z.S.; data curation, M.Z.S.; writing—original draft preparation, S.G. and M.Z.S.; writing—review and editing, S.G. and M.Z.S. All authors have read and agreed to the published version of the manuscript.

**Funding:** This research received no external funding.

**Institutional Review Board Statement:** Not applicable.

**Informed Consent Statement:** Not applicable.

**Data Availability Statement:** The data used in this study are publicly available at the World Bank Database.

**Acknowledgments:** The authors are grateful to the four anonymous referees and the journal editor for their helpful comments that helped improve the quality of this paper.

**Conflicts of Interest:** The authors declare no conflict of interest.

# Appendix A

**Table A1.** List of 147 developing countries.

| | | | |
|---|---|---|---|
| Afghanistan | South Sudan | Kazakhstan | Romania |
| Albania | Chile | Kenya | Russian Federation |
| Algeria | China | Kiribati | Rwanda |
| Angola | Colombia | Sri Lanka | Samoa |
| Antigua and Barbuda | Comoros | Liberia | Sao Tome and Principe |
| Argentina | Congo, Dem. Rep. | Libya | Saudi Arabia |
| Armenia | Costa Rica | Madagascar | Senegal |
| Aruba | Djibouti | Malawi | Serbia |
| Azerbaijan | Dominica | Malaysia | Seychelles |
| Bahamas | Dominican Republic | Maldives | Sierra Leone |
| Bahrain | Ecuador | Mali | Solomon Islands |
| Bangladesh | Egypt | Marshall Islands | Somalia |
| Barbados | El Salvador | Mauritania | South Africa |
| Belarus | Equatorial Guinea | Mauritius | St. Kitts and Nevis |
| Belize | Eritrea | Mexico | St. Lucia |
| Benin | Eswatini | Moldova | St. Vincent and the Grenadines |
| Bhutan | Ethiopia | Mongolia | Sudan |
| Bolivia | Fiji | Montenegro | Suriname |
| Bosnia and Herzegovina | Gabon | Morocco | Syria |
| Botswana | The Gambia | Mozambique | Tajikistan |
| Brazil | North Macedonia | Namibia | Tanzania |
| Brunei Darussalam | Georgia | Nauru | Thailand |
| Bulgaria | Ghana | Nepal | Togo |
| Burkina Faso | Grenada | Nicaragua | Tonga |
| Burundi | Guatemala | Niger | Trinidad and Tobago |
| Cabo Verde | Guinea-Bissau | Oman | Turkey |
| Cambodia | Guyana | Panama | Turkmenistan |
| Cameroon | Haiti | Pakistan | Tuvalu |
| Central African Republic | Honduras | Palau | Tunisia |
| Chad | India | Papua New Guinea | Uganda |
| Kosovo | Indonesia | Paraguay | Ukraine |
| Kuwait | Iran | Peru | Uruguay |
| Kyrgyzstan | Iraq | Philippines | Uzbekistan |
| Laos | Jamaica | Poland | Vanuatu |
| Lebanon | Jordan | Nigeria | Vietnam |
| Lesotho | Guinea | Qatar | Venezuela |
| Yemen | Zambia | Zimbabwe | |

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
