# Peer review of "Environmental Effects of Commodity Trade vs. Service Trade in Developing Countries"

_commodities, doi:10.3390/commodities1020008_

Round 1

Reviewer 1 Report

Dear Authors,

The aim of the submitted paper is to identify the effects of commodity trade and service trade on carbon dioxide emissions in developing countries. Such research is helpful and significant for policy makers in such a time when climate change imposes..

Point 1, the Abstract should begin with a background statement and end with meaningful statement so that the structure can be set completely.

Point 2, the second paragraph of Introduction said the ambiguities in current literature and empirical evidence,but we can not get the detailed information, can you depict or summarize it from the aboving statements? What’s more, the Introduction should contain a section that summarises the shortcomings of the existing literatures, but where does it show in the text?

Point 3. The contents of literatures in Table 1 are a bit of redundancy. It is recommended to simplify the contents and just emphasise the key points.

Point 4, Page 4 line110-157. These paragraphs of the literature review appear to be a superimposition of several specific literatures, and it is recommended that the relationship of the literature be reorganised according to the differences in the content, methodology, data and scale of the literatures in the study.

Point 5, Page 6. The 3.3 Estimation Techniques part should add the Hausman test presented in tabular form instead of pure written narrative.

Point 6, Page 9. The conclusions in the 5. Summary and Conclusions part should be further refined and optimized, and attention should be paid to the standard of language writing.

Author Response

Dear Reviewer,
Thank you for the opportunity to revise and resubmit our work. The comments from the four anonymous referees were extremely thoughtful and have helped improve the quality of our paper. In the attached document, we provide a point-by-point response to the comments. We first summarize the comments in black and then report our responses in blue. Accordingly, we have highlighted all changes in the revised manuscript in blue font.

Reviewer 2 Report

Abstract: Please write the end of the abstract the contribution of your study.

Introduction: Globalization's impact on growth also discuss in the background of the study. At the end of the study, please write the main objective of this study. Also, discuss the research Gap.

LR. I will suggest that the researcher discuss the LR based on the study during the CVOID as well. How much impact on the trade? 

Methodology Part ok, well explain.

Data analysis: why have you used the R software, not others? Please justify.

 Discussion: Based on your study finding, please suggest a few policies.

Author Response

(The authors gave the same response as above.)

Reviewer 3 Report

Abstract: What are differential effects?

Conclusion:

"I see that the goal was to find if decomposition of overall trade into commodity and service trade would have a heterogeneous impact on the environmental quality"

That is not clear, what is the goal? How do you define heterogenous impact? 

"We accomplish this goal by using total trade, commodity trade (using merchandize trade as % of GDP as a proxy), and service trade as explanatory variables; and CO2 emissions (proxy for environmental quality) as dependent variables respectively"

The use of trade, as % and separately service trade and commodity trade is not feasible.

" The results show that increase in trade has negative impacts on the environment, which comes from commodity trade rather than service trade"

What is the novelty of the findings?

"Appropriate policies are necessary to control gas emissions from manufacturing and industrial sectors which make up most of the commodity trade. To do so, it is imperative to first identify the commodities that are exported by the developing countries to the developed countries." 

Identification of all commodities is not possible. How you will choose the commodities you want to use for calculation?

Conclusion: You shouldn't put the references in the conclusion, just your own findings

Author Response

(The authors gave the same response as above.)

Reviewer 4 Report

The paper analyzes the impacts of trade openness on CO2 emissions from developing countries, decomposing the effects into openness, goods and services. The topic has been analyzed in the literature, but the paper provides some original standpoints. However, there are formal issues that have to be dealt with, and the model specification probably has to be reviewed or, at least, better explained, for the results to be sound.

Decomposition of the effects of trade in scale, technique and composition effects is mentioned both in lines 38-39 and 41-42

Line 47: suggest to try “public bad” instead of “public good”

Line 62: the word “accordingly” is confusing here and the idea should be explained better. The argument in the previous sentence is based on specialization production, but reference [16] is about efforts and effects of climate change.

Line 72: typo (“trade is services”)

Line 94: better not advance results in the introduction, delete “The results support these hypotheses”

Line 175: it should be specified the scope of CO2 emissions included in the World Bank dataset, instead of forcing the reader to do the job. Is it just CO2 coming from productive sectors or does it include as well the domestic sector? Does it include national transport? International transport? In any case, for the clarification of results, it would be advisable to express where CO2 emissions from international trade is included in the international statistics.

Lines 182 and 201: the large value of a variable cannot be taken as a reason to convert the variable to log form. Log transformation must be based on theory, and have significant consequences in the interpretation of results

Lines 195-198: the criteria actually used in the paper should be clarified

Table 3 should be reviewed: it makes no sense to have 9 decimal positions for figures

Table 3 should be reviewed: labor should be defined in the text before the estimation is performed (not only in Table 2). Now is defined in lines 278-285, but the definition is not completely satisfactory. “The labor force of population” should be clarified; are we talking about active population? Working population? Why one and not the other? Why labor is included in the model? The explanation is not satisfactory either, since the explained effect should be in the GDP growth variable. It is important to clarify this point, in order to trust the validity of the model

Table 3 should be reviewed: maximum figure for commodity trade (975.78) seems a typo

Lines 207-211: please explain Hausman procedure better

Lines 260-265: the explanation for “capital” variable is not satisfactory, it should be made in an independent way to trade since it is assumed to be an independent variable.

Lines 266-270: the explanation for “GDP” variable is not satisfactory, since the variable reflects growth of GDP (hence, the results refer to the growth of the growth of GDP). It is not well explained whether a positive or a negative sign is expected. Most of the literature is focused on a positive sign.

Lines 271-277: the explanation for “FDI” variable is not satisfactory. First, the author(s) talk about remittances, why? Second, FDI will likely come from delocalization, fleeing from more strict environmental regulations, a negative sign is against the effects described in the literature. In order to clarify, a deeper literature analysis should be carried out, as well as, possibly, a better model specification.

The conclusions should be modified consistently with the suggested changes.

Author Response

(The authors gave the same response as above.)

Round 2

Reviewer 3 Report

N/A

Reviewer 4 Report

The results are interesting and somewhat novel. My main advice in order to improve this reearch line would be to go deeper in the GDP influence, the Kuznets hypothesis might be an explanation for some of the results but it is not fully checked in the paper